# Mask Frozen-DETR: High Quality Instance Segmentation with One GPU

## Abstract

In this paper, we aim to study how to build a strong instance segmenter with minimal training time and GPUs, as opposed to the majority of current approaches that pursue more accurate instance segmenter by building more advanced frameworks at the cost of longer training time and higher GPU requirements. To achieve this, we introduce a simple and general framework, termed Mask Frozen-DETR, which can convert any existing DETR-based object detection model into a powerful instance segmentation model. Our method only requires training an additional lightweight mask network that predicts instance masks within the bounding boxes given by a frozen DETR-based object detector. Remarkably, our method outperforms the state-of-the-art instance segmentation method Mask DINO in terms of performance on the COCO test-dev split (55.3% vs. 54.7%) while being over $10\times$ times faster to train. Furthermore, all of our experiments can be trained using only one Tesla V100 GPU with 16 GB of memory, demonstrating the significant efficiency of our proposed framework.

## 1 Introduction

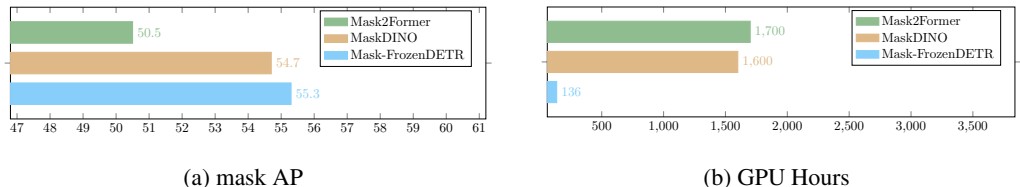

(a) mask AP                    (b) GPU Hours

Figure 1: Illustrating the comparison results with Mask2Former and Mask DINO on COCO instance segmentation task. (a) our Mask Frozen-DETR outperforms the previous SOTA Mask DINO by +0.6%. (b) our Mask Frozen-DETR speeds up the training by more than $10\times$ times compared to Mask DINO.

Instance segmentation is one of the most fundamental but difficult computer vision tasks requiring pixel-level localization and recognition in the given input image. Most advances in modern image instance segmentation methods are heavily influenced by the latest state-of-the-art 2D object detection systems. For example, two representative leading instance segmentation approaches, including Cascade Mask R-CNN (Cai & Vasconcelos, 2019) and Mask DINO (Li et al., 2022), are built by adding a parallel segmentation branch to the strong object detection systems, specifically Cascade R-CNN (Cai & Vasconcelos, 2018) and DINO (Zhang et al., 2022).

Despite the convergence of deep neural network architectures between object detection and instance segmentation, most existing efforts still need independent training based on supervision signals of different granularity, i.e., bounding boxes vs. instance masks. Training modern instance segmentation models from scratch is resource-intensive and time-consuming. For example, Using ResNet-50 (He et al., 2016) and Swin-L (Liu et al., 2021) as backbone networks, Mask2Former (Cheng et al., 2022a) requires over $500\times$ and $1700\times$ V100 GPU hours for training, respectively.

We show that the existing DETR-based object detection models can be efficiently converted into strong instance segmentation models, unlike the previous efforts that train the instance segmentation models from scratch. We start from the recent powerful 2D object detection models, i.e.,

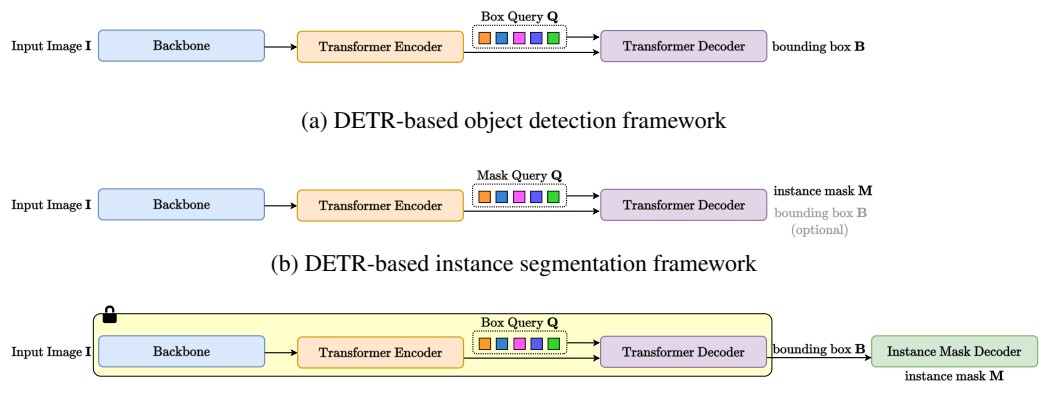

Figure 2: Illustrating the overall pipelines of DETR-based object detectors (1-st row), instance segmenters (2-ed row) based on DETR, and the proposed approach. Instead of training the instance segmentation models from scratch, we propose a Mask Frozen-DETR that uses a frozen DETR-based object detector to generate the bounding boxes and then trains a mask head to produce instance segmentation masks.

$\mathcal{H}$-DETR (Jia et al., 2022) and DINO-DETR (Zhang et al., 2022). We propose two key innovations: (i) design a light-weight instance segmentation network that effectively uses the output of a frozen DETR-based object detector, including the object query and the encoder feature map, to predict instance masks, and (ii) demonstrate the high training efficiency of our approach compared to the previous state-of-the-art instance segmentation approaches while achieving competitive performance under different model scales.

We conduct comprehensive comparison experiments on the COCO (Lin et al., 2014) instance segmentation benchmark to verify the effectiveness of our approach. Our approach achieves strong results with a very short learning time. For instance, with only $6\times$ training epochs, our approach slightly surpasses the state-of-the-art method Mask DINO. Remarkably, the entire training process of our approach takes less than $140\times$ GPU hours, while Mask DINO takes nearly $1600\times$ GPU hours. Consequently, our approach improves the training efficiency by more than $10\times$. We hope our simple approach can enable broader research communities to contribute to advancing stronger instance segmentation models.

## 2 RELATED WORK

**Object Detection.** Object detection is a fundamental research area that has produced a lot of excellent work, such as Faster R-CNN (Ren et al., 2015), Cascade R-CNN (Cai & Vasconcelos, 2018), YOLO (Redmon et al., 2016), DETR (Carion et al., 2020), and Deformable DETR (Zhu et al., 2020). Recently, most of the exciting progress in object detection mainly comes from the developments of various DETR-based approaches, including DINO-DETR (Zhang et al., 2022) and $\mathcal{H}$-DETR (Jia et al., 2022). The current state-of-the-art methods (Wang et al., 2022; Lin et al., 2023; Ma et al., 2023; Liang et al., 2022; He et al., 2022) are also built based on them. In general, we can easily access the weights of many DETR-based object detection models as most of them are open-sourced. We show that our approach can be extended to these modern DETR-based detectors easily and achieves strong performance while being more than $10\times$ faster to train by exploiting the off-the-shelf pre-trained weights on object detection tasks.

**Instance Segmentation.** Instance segmentation is a computer vision task that requires an algorithm to assign a pixel-level or point-level mask with a category label for each instance of interest in an image, video or point cloud. Most existing methods follow the R-CNN (Girshick et al., 2014) paradigm, which first detects objects and then segments them. For example, Mask R-CNN (He et al., 2017) extends Faster R-CNN (Ren et al., 2015) with a fully convolutional mask head, Casacde Mask R-CNN combines Casacde R-CNN (Cai & Vasconcelos, 2018) with Mask R-CNN, and HTC (Chen et al., 2019) improves the performance with interleaved execution and mask information flow. Some recent methods propose more concise designs such as SOLO (Wang et al., 2020) that segments objects by locations without bounding boxes or embedding learning, QueryInst (Fang et al., 2021)

that performs end-to-end instance segmentation based on Sparse RCNN (Sun et al., 2021), Mask-Former (Cheng et al., 2021b) and Mask2Former(Cheng et al., 2022a) that use a simple mask classification based on DETR (Carion et al., 2020), and Mask-DINO (Li et al., 2022) that extends DINO by adding a mask prediction branch which supports all image segmentation tasks using query embeddings and pixel embeddings. Moreover, the very recent Mask3D (Schult et al., 2022) and SP-Former (Sun et al., 2022) have built state-of-the-art 3D instance segmentation systems following the design of Mask2Former.

**Discussion.** Most of the existing efforts train the instance segmentation models from scratch without using the off-the-shelf object detection model weights, thus requiring very expensive training. Our approach uses the weights of frozen DETR-based models and introduces a very efficient instance segmentation head design with high training efficiency. Figure 2 illustrates the differences between the existing methods and our approach. For example, the very recent state-of-the-art Mask DINO is trained from scratch while we simply freeze the existing object detector and train a very light instance mask decoder. We empirically show the great advantages of our approach on the COCO instance segmentation task with experiments under various settings. Notably, our approach sets new records on the challenging COCO instance segmentation task while accerating the training speed by more than $10\times$.

# 3 OUR APPROACH

## 3.1 BASELINE SETUP

We use a strong object detector $\mathcal{H}$-DETR+ResNet50 with AP=52.2 as our baseline for the following ablation experiments and report the results based on stronger $\mathcal{H}$-DETR+Swin-L and DINO-DETR+FocalNet-L for the system-level comparisons. The entire $\mathcal{H}$-DETR+ResNet50 model is pre-trained on Object365 (Shao et al., 2019) and then fine-tuned on COCO for higher performance.

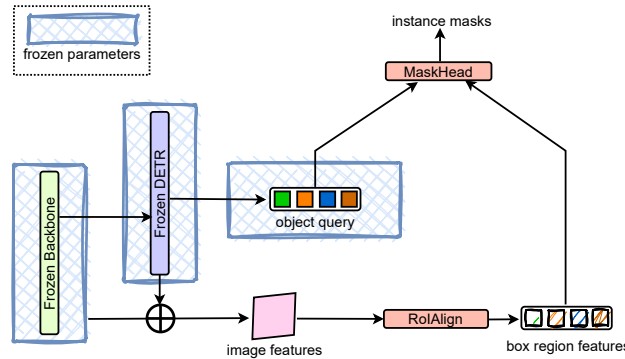

Figure 3: Mask Frozen-DETR baseline: both RoIAlign and MaskHead are non-parametric operations.

To build a simple baseline for instance segmentation without extra training, we first sort the object queries output by the last Transformer decoder layer of the object detection model according to the decreasing order of their classification scores, and select the top $\sim$100 object queries for mask prediction. Then we multiply the object queries $\{\mathbf{q}_i | \mathbf{q}_i \in \mathbb{R}^{\mathsf{d}}\}_{i=1}^{N}$ and the image features $\mathbf{F} \in \mathbb{R}^{\frac{\mathsf{HW}}{16} \times \mathsf{d}}$ at 1/4-resolution to get the instance segmentation masks as follows:

$$\mathbf{F} = \mathbf{C}_1 + \text{interpolate}(\mathbf{E}),$$
$$\mathbf{M}_i = \text{interpolate}(\text{reshape}(\text{Sigmoid}(\mathbf{q}_i \mathbf{F}^{\top}))), \tag{1}$$

where $\mathbf{C}_1 \in \mathbb{R}^{\frac{\mathsf{HW}}{16} \times \mathsf{d}}$ represents the feature map output by the first stage of the backbone. $\mathbf{E} \in \mathbb{R}^{\frac{\mathsf{HW}}{64} \times \mathsf{d}}$ represents the 1/8-resolution feature map from the Transformer encoder. H, W, and d represent the image height, image width, and feature hidden dimension, respectively. $\mathbf{M}_i \in \mathbb{R}^{\mathsf{HW}}$ represents the final predicted probability mask with the same resolution as the input image.

Next, we compute the confidence scores that reflect the quality of masks following:

$$\mathbf{s}_i = \mathbf{c}_i \times \frac{\text{sum}(\mathbf{M}_i[\mathbf{M}_i > 0.5])}{\text{sum}([\mathbf{M}_i > 0.5])}, \tag{2}$$

where $\mathbf{c}_i$ represents the classification score associated with the $i$-th object query predicted by the last Transformer decoder layer of the object detection model.

Table 1: **Effect of each factor within our baseline that requires no training.** RoIAlign: use RoIAlign to pool the region features according to the predicted boxes. 1/4 feat.: fuse the 1/4-resolution feature maps output by the stage-1 of backbone with the up-sampled 1/4-resolution feature maps output by the Transformer encoder. We use ▲ to mark the additional increased number of parameters and FLOPs in all the following tables.

| RoIAlign | 1/4 feat. | #FLOPs▲ | #params▲ | $AP^{mask}$ | $AP^{mask}_{50}$ | $AP^{mask}_{75}$ | $AP^{mask}_{S}$ | $AP^{mask}_{M}$ | $AP^{mask}_{L}$ |
|---|---|---|---|---|---|---|---|---|---|
| ✗ | ✗ | 1.49 G | 0.0 M | 0.0 | 0.1 | 0.0 | 0.0 | 0.0 | 0.1 |
| ✓ | ✗ | 0.53 G | 0.0 M | 4.4 | 14.8 | 1.3 | 2.7 | 4.6 | 7.5 |
| ✗ | ✓ | 1.50 G | 0.0 M | 0.0 | 0.1 | 0.0 | 0.0 | 0.0 | 0.1 |
| ✓ | ✓ | 0.54 G | 0.0 M | 5.1 | 17.0 | 1.5 | 3.2 | 4.9 | 8.4 |

We illustrate the modifications when using RoIAlign operation (He et al., 2017) as follows. Instead of using the entire image features $\mathbf{F} \in \mathbb{R}^{\frac{HW}{16} \times d}$, we use the RoIAlign to gather the region feature maps located within the predicted bounding boxes:

$$\mathbf{R}_i = \text{reshape}(\text{RoIAlign}(\text{reshape}(\mathbf{F}), \mathbf{b}_i)), \quad (3)$$

where $\mathbf{R}_i \in \mathbb{R}^{hw \times d}$, $\mathbf{b}_i$ represents the predicted bounding box of $\mathbf{q}_i$. We set h and w as 32 by default. Then we compute the instance segmentation masks as follows:

$$\mathbf{M}^r_i = \text{paste}(\text{interpolate}(\text{reshape}(\text{Sigmoid}(\mathbf{q}_i \mathbf{R}^\top_i)))), \quad (4)$$

where we first reshape and interpolate the predicted regional instance masks to be the same size as the real bounding box size in the original image and then paste the resized ones to an empty instance mask of the same size as the original image. We compute the confidence scores based on $\mathbf{M}^r_i$ following a similar manner. We illustrate the overall pipeline in Figure 3

**Results.** Table 1 shows the comparison results on the effect of fusing the 1/4-resolution feature maps output by the first stage of the backbone and using the RoIAlign to constrain the computation focus only within the predicted bounding boxes. According to the reported results, we find that directly multiplying the object queries with the image feature maps performs very poorly, i.e., the best setting achieves a mask AP score 5.1%. Notably, we also report the additional increased computational cost and number of parameters in all ablation experiments by default.

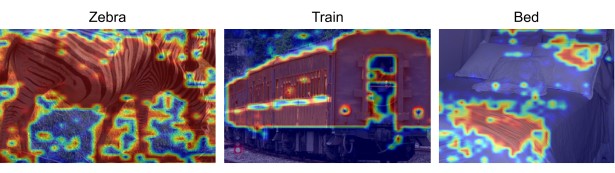

Figure 4: Coarse instance segmentation with the Mask frozen-DETR baseline. We visualize the predicted probability maps and the color indicates the confidence scores: red for high and blue for low.

We aim to make minimal modifications to boost the segmentation performance based on the above baseline setting. We show how to improve the system from three main aspects, including image feature encoder design, box region feature encoder design, and query feature encoder design, in the following discussions. We conduct all the following ablation experiments by freezing the entire object detection network and only fine-tuning the additional introduced parameters for ∼6 epochs on COCO instance segmentation task.

**Experiment Setup.** We use AdamW (Loshchilov & Hutter, 2017) optimizer with an initial learning rate $1.5 \times 10^{-4}$, $\beta_1 = 0.9$, $\beta_2 = 0.999$ and a weight decay of $5 \times 10^{-5}$ is employed. We train the models for $88,500$ iterations (i.e. 6 epochs), and divide the learning rate by 10 at 0.9 and 0.95 fractions of the total number of the training iterations. We adhere to the data pre-processing approach outlined in Deformable DETR (Zhu et al., 2020). Our experiments employ a batch size of 8, utilizing V100 GPUs equipped with 16GB of memory. Remarkably, we also present results achieved with a significantly reduced batch size of 2, which not only yields superior outcomes but also demands the utilization of only one V100 GPU. We report a set of COCO metrics including AP, $AP_{50}$, $AP_{75}$, $AP_S$, $AP_M$ and $AP_L$.

### 3.2 IMAGE FEATURE ENCODER

We first study how to improve the previous baseline setting by introducing a trainable image feature encoder to transform the image feature maps into a more suitable feature space for instance segmentation tasks. Figure 5 shows the overall pipeline. We apply the image feature encoder on feature map $\mathbf{E}$ from the Transformer encoder. We simply modify Equation 1 as follows:

Table 2: Effect of image feature encoder design.

| block type | # layers | #FLOPs▲ | #params▲ | $AP^{mask}$ | $AP_{50}^{mask}$ | $AP_{75}^{mask}$ | $AP_S^{mask}$ | $AP_M^{mask}$ | $AP_L^{mask}$ |
|---|---|---|---|---|---|---|---|---|---|
| None | 0 | 0.54 G | 0.0 M | 5.1 | 17.0 | 1.5 | 3.2 | 4.9 | 8.4 |
| deformable. | 1 | 14.71 G | 0.76 M | 30.8 | 58.2 | 29.5 | 13.5 | 32.7 | 49.0 |
| deformable. | 2 | 28.88 G | 1.51 M | 32.9 | 60.0 | 32.4 | 14.9 | 35.3 | 52.1 |
| deformable. | 3 | 43.04 G | 2.27 M | 33.7 | 60.5 | 33.7 | 15.1 | 36.3 | 53.4 |
| Swin Trans. | 1 | 14.20 G | 0.80 M | 30.0 | 58.0 | 28.0 | 13.8 | 32.3 | 46.3 |
| Swin Trans. | 2 | 27.86 G | 1.59 M | 31.6 | 59.2 | 30.4 | 14.5 | 34.2 | 48.6 |
| Swin Trans. | 3 | 41.51 G | 2.39 M | 32.2 | 59.6 | 31.3 | 14.9 | 34.8 | 49.5 |
| ConvNext. | 1 | 8.12 G | 0.54 M | 27.5 | 55.9 | 24.4 | 12.7 | 29.8 | 42.1 |
| ConvNext. | 2 | 15.70 G | 1.08 M | 30.0 | 58.1 | 27.8 | 13.9 | 32.4 | 46.0 |
| ConvNext. | 3 | 23.27 G | 1.62 M | 30.7 | 58.5 | 28.9 | 14.3 | 33.2 | 46.9 |

$$\mathbf{F} = \mathbf{C}_1 + \text{interpolate}(\mathcal{F}_e(\mathbf{E})), \tag{5}$$

where the $\mathcal{F}_e(\cdot)$ represents the image feature encoder that refines the image feature map for all object queries simultaneously. We study the following three kinds of modern convolution or transformer blocks.

**Deformable encoder block (Zhu et al., 2020).** We follow the multi-scale deformable encoder design and stack multiple multi-scale deformable encoder blocks to enhance the multi-scale feature map $\mathbf{E}$ following:

$$\mathbf{E} = [\mathbf{E}_1, \mathbf{E}_2, \mathbf{E}_3, \mathbf{E}_4],$$
$$\mathcal{F}_e(\mathbf{E}) = \text{MultiScaleDeformableEnc}([\mathbf{E}_1, \mathbf{E}_2, \mathbf{E}_3, \mathbf{E}_4]), \tag{6}$$

where $\mathbf{E}_1$, $\mathbf{E}_2$, $\mathbf{E}_3$, and $\mathbf{E}_4$ represent the feature maps of different scales from the Transformer encoder of the object detection system. Each multi-scale deformable encoder block is implemented with MSDeformAttn → LayerNorm → FFN → LayerNorm. FFN is implemented as Linear → GELU → Linear by default. MSDeformAttn represents the multi-scale deformable attention.

**Swin Transformer encoder block (Liu et al., 2021)** We follow the Swin Transformer to apply a stack of multiple Swin Transformer blocks on the feature map $\mathbf{E}_1$ with highest resolution as follows:

$$\mathcal{F}_e(\mathbf{E}_1) = \text{SwinTransformerEnc}(\mathbf{E}_1), \tag{7}$$

where each Swin Transformer block is implemented as LayerNorm → W-MSA → LayerNorm → FFN. W-MSA represents the window multi-head self-attention operation. We apply shifted W-MSA in the successive block to propogate information across windows following Liu et al. (2021).

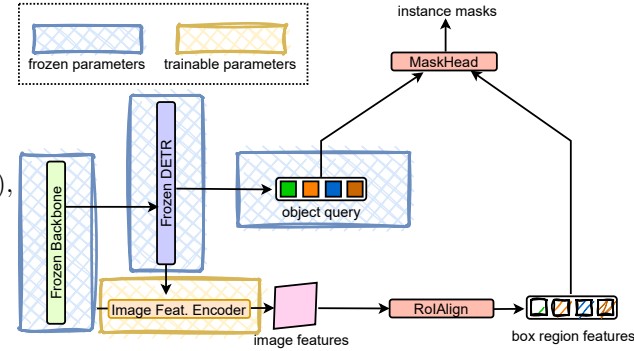

Figure 5: Add image feature encoder to Mask Frozen-DETR. We insert an additional image feature encoder to enhance the image feature maps.

**ConvNext encoder block (Liu et al., 2022)** We follow the ConvNext to apply the proposed combination of large-kernel convolution and inverted bottleneck on the Transformer encoder feature map $\mathbf{E}_1$ following:

$$\mathcal{F}_e(\mathbf{E}_1) = \text{ConvNextBlock}(\mathbf{E}_1). \tag{8}$$

where each ConvNext block is implemented as DWC → LayerNorm → FFN. DWC represents a depth-wise convolution with large kernel size, i.e., $7 \times 7$.

**Results.** In Table 2, we compare different choices of the image feature encoder architecture design. We observe that: (i) All three image feature encoder implementations improve the instance segmentation performance. (ii) More image feature encoder layers leads to better performance, e.g., $3\times$ deformable encoder layers: AP=32.9% vs. $1\times$ deformable encoder layer: AP=30.8%. (iii) Under similar computation budget, Deformable encoder block performs better, e.g., $2\times$ deformable encoder layers: AP=32.9%/FLOPs=28.88G vs. $2\times$ Swin transformer encoder layers: AP=31.6%/FLOPs=27.86G vs. $3\times$ ConvNext encoder layers: AP=30.7%/FLOPs=23.27G. We use $2\times$ deformable encoder layers as the image feature encoder in the following experiments as it has a better trade-off between performance and computational cost.

Table 3: Effect of box feature encoder design.

| block type | # layers | #FLOPs▲ | #params▲ | $AP^{mask}$ | $AP_{50}^{mask}$ | $AP_{75}^{mask}$ | $AP_{S}^{mask}$ | $AP_{M}^{mask}$ | $AP_{L}^{mask}$ |
|---|---|---|---|---|---|---|---|---|---|
| None | 0 | 28.88 G | 1.51 M | 32.9 | 60.0 | 32.4 | 14.9 | 35.3 | 52.1 |
| deformable. | 1 | 98.55 G | 2.46 M | 44.4 | 67.6 | 48.0 | 24.5 | 47.7 | 62.8 |
| deformable. | 2 | 168.23 G | 3.14 M | 44.9 | 67.7 | 48.6 | 24.9 | 48.2 | 63.4 |
| deformable. | 3 | 237.91 G | 3.82 M | 45.1 | 67.8 | 48.9 | 25.2 | 48.4 | 63.7 |
| Swin Trans. | 1 | 112.84 G | 2.31 M | 42.5 | 66.3 | 45.4 | 23.0 | 45.5 | 61.2 |
| Swin Trans. | 2 | 196.81 G | 3.10 M | 43.7 | 66.9 | 46.8 | 23.8 | 46.8 | 62.3 |
| Swin Trans. | 3 | 280.77 G | 3.89 M | 44.3 | 67.3 | 47.6 | 24.4 | 47.6 | 62.8 |
| ConvNext. | 1 | 83.90 G | 2.05 M | 40.9 | 65.8 | 43.4 | 21.4 | 43.8 | 59.9 |
| ConvNext. | 2 | 138.92 G | 2.59 M | 43.0 | 66.9 | 46.0 | 23.4 | 46.1 | 61.7 |
| ConvNext. | 3 | 193.95 G | 3.13 M | 43.7 | 67.2 | 47.1 | 23.9 | 47.0 | 62.1 |

Table 4: Effect of hidden dimension after channel mapper.

| # hidden dim. | #FLOPs▲ | #params▲ | $AP^{mask}$ | $AP_{50}^{mask}$ | $AP_{75}^{mask}$ | $AP_{S}^{mask}$ | $AP_{M}^{mask}$ | $AP_{L}^{mask}$ |
|---|---|---|---|---|---|---|---|---|
| 256 | 168.23 G | 3.14 M | 44.9 | 67.7 | 48.6 | 24.9 | 48.2 | 63.4 |
| 128 | 66.60 G | 2.07 M | 44.7 | 67.6 | 48.2 | 24.8 | 48.1 | 63.2 |
| 96 | 50.76 G | 1.87 M | 44.6 | 67.5 | 48.2 | 24.7 | 47.9 | 63.2 |
| 64 | 39.11 G | 1.71 M | 44.5 | 67.5 | 48.0 | 24.7 | 47.9 | 63.0 |

## 3.3 BOX FEATURE ENCODER

Now we study the influence of improving the box region features with an additional box feature encoder design. We illustrate the modification in Figure 6. we simply apply additional transformation $\mathcal{F}_b$ on $\mathbf{R}_i$ before Equation 4 following:

$$\mathbf{R}_i = \mathcal{F}_b(\mathbf{R}_i), \qquad (9)$$

where we study different choices of the box feature encoder implementation following the study on the image feature encoder design.

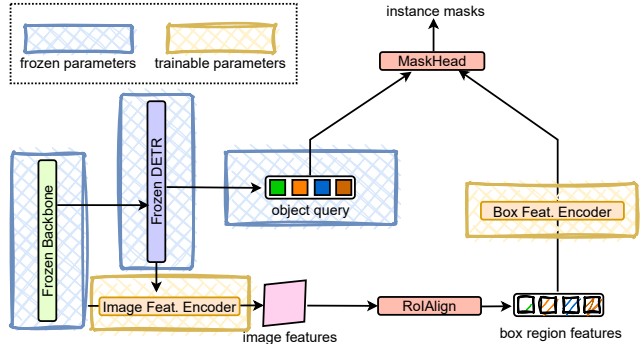

Figure 6: Add box feature encoder to Mask Frozen-DETR. We insert a feature encoder to enhance the bounding box region features.

**Channel mapper.** To build an efficient box feature encoder, we propose to use a channel mapper as a simple Linear layer to decrease the channel dimension of $\mathbf{F} \in \mathbb{R}^{\frac{HW}{16} \times d}$ and apply the box region feature encoder on the updated feature map with smaller channels.

**Results.** We compare the effect of box feature encoder architecture design in Table 3. We notice significant computational cost increase for all settings, mainly due to two reasons: (i) the large number of bounding box region features, i.e., 100 box predictions during inference; and (ii) the high-resolution of the bounding box feature map, i.e., $32 \times 32$. We study the impact of different resolutions of the region feature maps in the ablation experiments. We find similar conclusions as Table 2: all methods improve the performance, more encoder layers lead to better performance, and deformable encoder block performs the best. Therefore, we use deformable encoder blocks for the box feature encoder. To reduce the expensive computational cost, we use the channel mapper to decrease the hidden dimension. The results are in Table 4. We observe that lower hidden dimension reduces computational cost with little performance loss. Considering the trade-off between performance and overhead, we use 128 hidden dimensions for subsequent experiments.

## 3.4 QUERY FEATURE ENCODER

After studying the influence of adding the image feature encoder and box region feature encoder, we further investigate how to design a suitable query feature encoder to refine the object queries originally designed for detecting the boxes for instance segmentation tasks.

Table 5: Effect of each factor within the query feature encoder. O2O: object-to-object attention module. B2O: box-to-object attention module.

| FFN | O2O | B2O | #FLOPs▲ | #params▲ | $AP^{mask}$ | $AP_{50}^{mask}$ | $AP_{75}^{mask}$ | $AP_{S}^{mask}$ | $AP_{M}^{mask}$ | $AP_{L}^{mask}$ |
|---|---|---|---|---|---|---|---|---|---|---|
| ✗ | ✗ | ✗ | 66.60 G | 2.07 M | 44.7 | 67.6 | 48.2 | 24.8 | 48.1 | 63.2 |
| ✓ | ✗ | ✗ | 66.63 G | 2.33 M | 44.7 | 67.6 | 48.3 | 24.9 | 48.0 | 63.2 |
| ✓ | ✓ | ✗ | 66.65 G | 2.53 M | 43.9 | 67.1 | 47.3 | 24.4 | 47.3 | 61.8 |
| ✓ | ✗ | ✓ | 73.40 G | 2.46 M | 44.8 | 67.6 | 48.4 | 24.9 | 48.1 | 63.4 |

**Object-to-object attention.** The proximity of objects to each other may results in a situation where multiple instances lie within one bounding box. We add object-to-object attention to help object query representations distinguish instances. Specifically, we use multi-head self-attention mechanism to process the queries as follows:

$$[\mathbf{q}_1, \mathbf{q}_2, \cdots, \mathbf{q}_N] = \text{SelfAttention}([\mathbf{q}_1, \mathbf{q}_2, \cdots, \mathbf{q}_N]). \tag{10}$$

**Box-to-object attention.** In the frozen DETR-based object detector, the object queries are used to perform object detection and process the whole image feature instead of a box region feature. The discrepancies between the usage of object queries in the frozen detector and MaskHead may lead to sub-optimal segmentation results. Therefore, we introduce box-to-object attention to transform the queries and adapt them to the segmentation task as follows:

$$\mathbf{q}_i = \text{CrossAttention}(\mathbf{q}_i, \mathbf{R}_i), \tag{11}$$

where the object queries are updated by refering to the information in the box region feature.

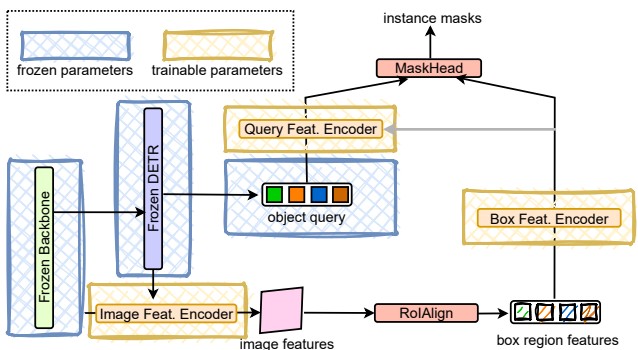

Figure 7: Add object query encoder to Frozen DETR. We apply a query feature encoder to enhance the object query representations. This is the complete framework of our Mask Frozen-DETR.

**FFN.** The feed forward network (FFN) block is a widely employed element in Transformers. It is usually integrated after an attention layer to transform individual tokens. In Table 5, we investigate the efficacy of this block in adjusting object query representations for segmentation.

**Results.** Table 5 shows the comparison results on the effect of different modifications to the query feature encoder architecture design. We find no performance improvement when only using FFN to enhance the object queries and a minor gain (+0.1) when using FFN and the box-to-object attention module. We think this is because the enhanced box features are already strong for instance segmentation, so using the original object query representations from the frozen detection model is enough. Moreover, using object-to-object attention reduces the performance, as the interaction between object queries might mix the semantic information of different objects. In general, we conclude that transforming the object queries is unnecessary and simply using the original ones to interact with the refined box region features achieves strong results. We only adopt FFN and the box-to-object attention design in the qualitative analysis experiments.

We also have boosted our system through additional enhancements, including mask loss on sampled pixel points, a refined mask scoring scheme, and a redesigned neck for backbone features. Further details are available in the supplementary materials.

## 4 COMPARISON WITH SOTA SYSTEMS

To compare our system with state-of-the-art instance segmentation methods, we construct a series of strong Mask Frozen-DETR models based on different Frozen DETR-based detector weights. These include Mask Frozen-$\mathcal{H}$-DETR + ResNet-50, which uses $\mathcal{H}$-DETR + ResNet-50 with $AP^{box}$ of 52.2%; Mask Frozen-$\mathcal{H}$-DETR + Swin-L, which uses $\mathcal{H}$-DETR + Swin-L with $AP^{box}$ of 62.3%; and Mask Frozen-DINO-DETR + Swin-L, which uses DINO-DETR + FocalNet-L with $AP^{box}$ of 63.2%.

Table 6: Comparison with SOTA instance segmentation methods on COCO val.

| method | backbone | #epochs | Object365 | $AP^{box}$ | $AP^{mask}$ | $AP_{50}^{mask}$ | $AP_{75}^{mask}$ | $AP_S^{mask}$ | $AP_M^{mask}$ | $AP_L^{mask}$ | GPU Hours |
|---|---|---|---|---|---|---|---|---|---|---|---|
| K-Net-N256 (Zhang et al., 2021) | R50 | 36 | ✗ | − | 38.6 | 60.9 | 41.0 | 19.1 | 42.0 | 57.7 | − |
| QueryInst (Fang et al., 2021) | Swin-L | 50 | ✗ | 56.1 | 48.9 | 74.0 | 53.9 | 30.8 | 52.6 | 68.3 | − |
| Mask2Former (Cheng et al., 2021a) | R50 | 50 | ✗ | − | 43.7 | − | − | 23.4 | 47.2 | 64.8 | 502 |
| Mask2Former (Cheng et al., 2021a) | Swin-L | 100 | ✗ | − | 50.1 | − | − | 29.9 | 53.9 | 72.1 | 1,700 |
| Mask DINO (Li et al., 2022) | R50 | 50 | ✗ | 50.5 | 46.0 | 68.9 | 50.3 | 26.0 | 49.3 | 65.5 | 1,404 |
| Mask DINO (Li et al., 2022) | Swin-L | 50 | ✗ | 58.3 | 52.1 | 76.5 | 57.6 | 32.9 | 55.4 | 72.5 | 2,400 |
| Mask Frozen-$\mathcal{H}$-DETR | R50 | 6 | ✗ | 49.9 | 44.1 | 66.2 | 47.8 | 24.4 | 47.0 | 62.6 | 49 |
| Mask Frozen-$\mathcal{H}$-DETR | Swin-L | 6 | ✗ | 59.1 | 51.9 | 75.8 | 57.2 | 31.6 | 55.1 | 71.6 | 179 |
| ViT-Adapter-L (Chen et al., 2022) | ViT-L | 8 | ✓ | 61.8 | 53.0 | − | − | − | − | − | 1,068 |
| Mask DINO (Li et al., 2022) | Swin-L | 24 | ✓ | − | 54.5 | − | − | − | − | − | 1,600 |
| Mask Frozen-$\mathcal{H}$-DETR | R50 | 6 | ✓ | 52.2 | 45.7 | 67.5 | 49.8 | 25.6 | 48.9 | 64.1 | 49 |
| Mask Frozen-$\mathcal{H}$-DETR | Swin-L | 6 | ✓ | 62.3 | 54.0 | 77.9 | 59.5 | 35.6 | 57.4 | 73.0 | 172 |
| Mask Frozen-DINO-DETR | FocalNet-L | 6 | ✓ | 63.2 | **54.9** | **78.9** | **60.8** | **37.2** | **58.4** | **72.9** | 136 |

Table 7: Comparison with SOTA instance segmentation methods on COCO test-dev.

| method | backbone | #epochs | Object365 | $AP^{box}$ | $AP^{mask}$ | $AP_{50}^{mask}$ | $AP_{75}^{mask}$ | $AP_S^{mask}$ | $AP_M^{mask}$ | $AP_L^{mask}$ |
|---|---|---|---|---|---|---|---|---|---|---|
| K-Net-N256 (Zhang et al., 2021) | R101 | 36 | ✗ | - | 40.6 | 63.3 | 43.7 | 18.8 | 43.3 | 59.0 |
| SOLQ (Dong et al., 2021) | Swin-L | 50 | ✗ | 56.5 | 46.7 | - | - | 29.2 | 50.1 | 60.9 |
| SOIT (Yu et al., 2022) | Swin-L | 36 | ✗ | 56.9 | 49.2 | 74.3 | 53.5 | 30.2 | 52.7 | 65.2 |
| QueryInst (Fang et al., 2021) | Swin-L | 50 | ✗ | 56.1 | 49.1 | 74.2 | 53.8 | 31.5 | 51.8 | 63.2 |
| Mask2Former (Cheng et al., 2021a) | Swin-L | 100 | ✗ | - | 50.5 | 74.9 | 54.9 | 29.1 | 53.8 | **71.2** |
| Mask DINO (Li et al., 2022) | Swin-L | 24 | ✓ | - | 54.7 | - | - | - | - | - |
| Mask Frozen-DINO-DETR | FocalNet-L | 6 | ✓ | 63.2 | **55.3** | **79.3** | **61.4** | **37.8** | **58.4** | 70.4 |

Table 6 presents the detailed comparison results on COCO `val` set. We can see that, with Object365 object detection pre-training, our approach surpasses the very recent state-of-the-art Mask-DINO by a clear margin (ours: $54.9\%$ vs. Mask DINO: $54.5\%$). This result is remarkable considering that the strong object detector DINO-DETR achieves even better object detection performance than the DINO-DETR + FocalNet-L that we use. The most important advantage of our approach is the significantly reduced training time, e.g., we can complete the training of DINO-DETR + FocalNet-L within $17\times$ **hours** while training a Mask DINO + Swin-L takes more than $8\times$ **days** when using $8\times$ V100 GPUs.

We also provide the detailed comparison results for Mask Frozen-$\mathcal{H}$-DETR + ResNet-50 and Mask Frozen-$\mathcal{H}$-DETR + Swin-L. In general, our approach achieves competitive results across various model sizes and different DETR-based frameworks. We further compare Mask Frozen-DINO-DETR to the state-of-the-art methods in instance segmentation on COCO test-dev in Table 7. Mask Frozen-DINO-DETR with FocalNet-L achieves an AP of $55.3\%$. This result surpasses the recent state-of-the-art method, Mask DINO, by +0.6 AP.

## 5 ABLATION EXPERIMENTS AND ANALYSIS

**Effect of batch size.** Table 8 shows the effect of batch size. We observe that reducing the batch size from 8 to 4 or 2 even brings slight improvements in performance (+0.2 AP). It is worth noting that our method can run on a single V100 GPU with 16G memory when the batch size is 2. This highlights the effectiveness and computational resource efficiency of our approach.

**Effect of training epochs.** Table 9 shows the effect of training epochs. We notice that doubling the number of training epochs only brings +0.3 gain in AP. This result suggests that our model achieves convergence promptly, which can effectively reduce the required training time.

Table 8: Effect of batch size.

| batch size | $AP^{mask}$ | $AP_{50}^{mask}$ | $AP_{75}^{mask}$ | $AP_S^{mask}$ | $AP_M^{mask}$ | $AP_L^{mask}$ |
|---|---|---|---|---|---|---|
| 2 | 45.9 | 67.7 | 50.0 | 25.7 | 49.0 | 64.2 |
| 4 | 45.9 | 67.6 | 49.9 | 25.8 | 49.0 | 64.3 |
| 8 | 45.7 | 67.5 | 49.8 | 25.6 | 48.9 | 64.1 |

Table 9: Effect of training epochs.

| epoch | $AP^{mask}$ | $AP_{50}^{mask}$ | $AP_{75}^{mask}$ | $AP_S^{mask}$ | $AP_M^{mask}$ | $AP_L^{mask}$ |
|---|---|---|---|---|---|---|
| 6 | 45.7 | 67.5 | 49.8 | 25.6 | 48.9 | 64.1 |
| 12 | 46.0 | 67.7 | 50.3 | 25.9 | 49.1 | 64.3 |

Table 10: Effect of RoIAlign output size.

| output size | #FLOPs▲ | #params▲ | $AP^{mask}$ | $AP_{50}^{mask}$ | $AP_{75}^{mask}$ | $AP_S^{mask}$ | $AP_M^{mask}$ | $AP_L^{mask}$ |
|---|---|---|---|---|---|---|---|---|
| $16 \times 16$ | 46.09 G | 2.97 M | 44.8 | 67.5 | 49.1 | 25.4 | 48.0 | 62.0 |
| $32 \times 32$ | 82.09 G | 3.26 M | 45.7 | 67.5 | 49.8 | 25.6 | 48.9 | 64.1 |
| $64 \times 64$ | 225.86 G | 4.44 M | 45.9 | 67.6 | 50.2 | 25.8 | 49.1 | 64.4 |

Table 11: Effect of fine-tuning the whole DETR (fine-tune) or only the transformer encoder & decoder within DETR (partial fine-tune). During fine-tuning, we set the learning rate of the original DETR parameters as $1/10$ of the ones of the additional new parameters.

| detector method | image feat. enc. | box feat. enc. | query feat. enc. | partial finetune | finetune | $AP^{mask}$ | GPU Hours |
|---|---|---|---|---|---|---|---|
| | ✓ | ✓ | ✓ | ✗ | ✗ | 45.7 | 49 |
| | ✗ | ✗ | ✗ | ✓ | ✗ | 43.8 | 92 |
| $\mathcal{H}$-DETR+R50 | ✗ | ✗ | ✗ | ✗ | ✓ | 43.9 | 99 |
| | ✓ | ✓ | ✓ | ✓ | ✗ | 45.6 | 100 |
| | ✓ | ✓ | ✓ | ✗ | ✓ | 46.0 | 108 |
| | ✓ | ✓ | ✓ | ✗ | ✗ | 54.0 | 172 |
| $\mathcal{H}$-DETR+Swin-L | ✓ | ✓ | ✓ | ✓ | ✗ | 54.1 | 406 |
| | ✓ | ✓ | ✓ | ✗ | ✓ | 54.1 | 532 |
| | ✓ | ✓ | ✓ | ✗ | ✗ | 54.9 | 136 |
| DINO-DETR+FocalNet | ✓ | ✓ | ✓ | ✓ | ✗ | 54.9 | 218 |
| | ✓ | ✓ | ✓ | ✗ | ✓ | 55.0 | 319 |

**Effect of RoIAlign output size.** Table 10 shows the influence of RoIAlign output size. We observe that (i) Enlarging the RoIAlign output size significantly increases GFLOPs. (ii) Increasing RoIAlign output size can improve instance segmentation performance. Specifically, we observe a 0.9 improvement in AP by increasing the output size from $16 \times 16$ to $32 \times 32$. (iii) Instance segmentation performance saturates beyond a RoIAlign output size of $32 \times 32$. Taking into account the trade-off between performance and computational cost, we set RoIAlign output size as $32 \times 32$ by default.

**Effect of fine-tuning DETR**: We further ablate the effect of fine-tuning the DETR-based object detector either entirely or partially in Table 11. Accordingly, we observe that (i) fine-tuning DETR brings consistent, albeit marginal, gains while significantly increasing the overall training GPU hours; (ii) our method achieves the best trade-off between performance and training cost.

Furthermore, we investigate the impact of several factors in the supplementary material, including but not limited to the usage of large scale jittering, the architectural design of the instance mask head, the depth of both the image feature encoder and the box feature encoder, and various other aspects.

## 6 CONCLUSION

In this work, we have presented the detailed techniques for converting an existing off-the-shelf DETR-based object detector into a strong instance segmentation model with minimal training time and resources. Our approach is remarkably simple yet effective. We verify the effectiveness of our approach by reporting state-of-the-art instance segmentation results while accelerating the training by more than $10\times$ times. We believe our simple approach can inspire more research on advancing the state-of-the-art in instance segmentation model design.

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

# A OTHER IMPROVEMENTS

Table 12: Effect of other improvements including sampled pixel supervision, mask scoring, and neck design.

| neck | samp. pixel sup. | mask scoring | #FLOPs▲ | #params▲ | $AP^{mask}$ | $AP_{50}^{mask}$ | $AP_{75}^{mask}$ | $AP_{S}^{mask}$ | $AP_{M}^{mask}$ | $AP_{L}^{mask}$ |
|---|---|---|---|---|---|---|---|---|---|---|
| ✗ | ✗ | ✗ | 73.40 G | 2.46 M | 44.8 | 67.6 | 48.4 | 24.9 | 48.1 | 63.4 |
| ✓ | ✗ | ✗ | 77.07 G | 2.53 M | 45.2 | 67.8 | 48.9 | 25.3 | 48.5 | 63.8 |
| ✓ | ✓ | ✗ | 77.07 G | 2.53 M | 45.3 | 67.8 | 49.2 | 25.4 | 48.6 | 64.1 |
| ✓ | ✓ | ✓ | 82.09 G | 3.26 M | 45.7 | 67.5 | 49.8 | 25.6 | 48.9 | 64.1 |

**Mask loss on sampled pixel points.** Inspired by implicit PointRend (Cheng et al., 2022b) that shows a segmentation model can be trained with its mask loss computed on $N$ sampled points instead of the entire mask, we compute the mask loss with sampled points in the the final loss calculation. Specifically, given number of points $N$, oversample ratio $k$ ($k > 1$) and importance sample ratio $\beta$ ($\beta \in [0, 1]$), we randomly sample $kN$ points from the output mask and select $\beta N$ most uncertain points from the sampled points. Then we randomly sample other $(1 - \beta)N$ points from the output mask and compute loss only on these $N$ points.

**Mask scoring.** Since the classification scores predicted by the frozen DETR cannot reflect the quality of segmentation masks, we introduce mask scoring (Huang et al., 2019) to our method to adjust the score, making it able to describe the quality of segmentation masks more precisely. Specifically, the mask scoring head takes the output mask and box region features as input and uses them to predict the iou score between the output mask and ground truth following:

$$\text{iou}_i = \text{MLP}(\text{Flatten}(\text{Conv}(\text{Cat}(\mathbf{M}_i, \mathbf{R}_i)))), \qquad (12)$$

where $\text{Cat}$, $\text{Conv}$ and $\text{MLP}$ refer to concatenation, covolution layers and multilayer perceptron, respectively. The iou score predicted by the mask scoring head is then used to adjust the classification score as follows:

$$\text{s}_i = \text{c}_i \text{iou}_i, \qquad (13)$$

where $\text{s}_i$ is the confidence score of the output mask.

**Neck for backbone feature.** The feature map output by the first stage of backbone $\mathbf{C}_1 \in \mathbb{R}^{\frac{HW}{16} \times d}$ may not contain sufficient semantic information for accurate instance segmentation. Therefore, we introduce a simple neck block to encode more semantic information into the high resolution feature map $\mathbf{C}_1$. The neck block can be described using the following formula:

$$\mathbf{C}_1 = \text{GN}(\text{PWConv}(\mathbf{C}_1)), \qquad (14)$$

where $\text{GN}$ and $\text{PWConv}$ refer to group normalization (Wu & He, 2018) and point-wise convolution, respectively.

**Results.** In Table 12, we attempt to further improve the results by integrating a neck block design, using sampled pixel supervision, and using mask scoring. We observe that all three designs bring consistent gains in AP scores. For example, using the neck block improves AP from 44.8% to 45.2% and using mask scoring improves AP from 45.3% to 45.7%. Notably, while sampled pixel supervision reduces the number of the points for training supervisions by 75%, it still brings a slight gain in AP (+0.1%). Therefore, we use all three designs in the following experiments.

## B  ADDITIONAL ABLATION EXPERIMENTS AND ANALYSIS

**Large scale jittering.** Table 13 shows the effect of large scale jittering. We observe that using large scale jittering achieves a 0.3 AP improvement and increase the training GPU hours by 28.6%. In light of the trade-off between training time and performance, we do not utilize large scale jittering in other experiments.

Table 13: Effect of large scale jittering.

| LSJ | GPU Hour | $AP^{mask}$ | $AP_{50}^{mask}$ | $AP_{75}^{mask}$ | $AP_{S}^{mask}$ | $AP_{M}^{mask}$ | $AP_{L}^{mask}$ |
|---|---|---|---|---|---|---|---|
| ✗ | 49 | 45.7 | 67.5 | 49.8 | 25.6 | 48.9 | 64.1 |
| ✓ | 63 | 46.0 | 67.7 | 50.1 | 26.1 | 49.1 | 64.0 |

**Instance mask head design.** We compare the effect of mask head design in Table 14. Compared with segmenter head (Strudel et al., 2021) that contains linear projection and normalization, our simple dot product design achieves comparable AP scores with lower FLOPs and fewer number of parameters.

Table 14:  Effect of instance mask head design.

| mask head | #FLOPs▲ | #params▲ | $AP^{mask}$ | $AP_{50}^{mask}$ | $AP_{75}^{mask}$ | $AP_{S}^{mask}$ | $AP_{M}^{mask}$ | $AP_{L}^{mask}$ |
|---|---|---|---|---|---|---|---|---|
| dot product | 82.09 G | 3.26 M | 45.7 | 67.5 | 49.8 | 25.6 | 48.9 | 64.1 |
| segmenter head | 83.77 G | 3.29 M | 45.7 | 67.6 | 49.9 | 25.6 | 48.9 | 64.1 |

**Layer index of the encoder feature map.** We compare the encoder feature map **E** from different layers of the Transformer encoder in Table 15. We notice that using the encoder feature map from shallower layers outperforms using that from the last layer of the encoder. We think this is because the feature maps from the last layer of the encoder contain task-specific information relevant to detection, while feature maps from shallower layers contain more generalized object information that may facilitate segmentation. Therefore, we use the encoder feature map from layer#4 by default.

Table 15:  Layer index of the encoder feature map **E**.

| layer# | $AP^{mask}$ | $AP_{50}^{mask}$ | $AP_{75}^{mask}$ | $AP_{S}^{mask}$ | $AP_{M}^{mask}$ | $AP_{L}^{mask}$ |
|---|---|---|---|---|---|---|
| 2 | 45.7 | 67.5 | 49.8 | 25.7 | 48.9 | 64.0 |
| 4 | 45.7 | 67.5 | 49.8 | 25.6 | 48.9 | 64.1 |
| 6 | 45.5 | 67.5 | 49.3 | 25.6 | 48.7 | 63.7 |

**Effect of the depth of the image feature encoder and the box feature encoder** Table 16 shows the influence of the depth of the image feature encoder and the box feature encoder on Mask Frozen-DINO-DETR with FocalNet-L as backbone. We observe that: (i) The instance segmentation performance of Frozen-DINO-DETR reaches saturation when the depth of the box feature encoder is 2, and further increasing its depth does not result in a performance gain. (ii) Increasing the depth of image feature encoder from 2 to 3 leads to a 0.1 increase in AP. Nevertheless, the performance saturates when the depth of image feature encoder is 3. Given these findings, we select the depth of the image feature encoder to be 3 and the depth of the box feature encoder to be 2 for the comparisons with state-of-the-art instance segmentation methods on the COCO test-dev.

## C  QUALITATIVE RESULTS

Figure 8 shows the instance segmentation probability maps based on our approach. We notice that the probability maps precisely capture the object boundaries, which supports the strong performance of our approach on instance segmentation tasks.

Table 16: Effect of the depth of the image feature encoder and the box feature encoder on DINO + FocalNet-L.

| # img. enc. layers | # box enc. layers | #FLOPs▲ | #params▲ | $AP^{mask}$ | $AP^{mask}_{50}$ | $AP^{mask}_{75}$ | $AP^{mask}_S$ | $AP^{mask}_M$ | $AP^{mask}_L$ |
|---|---|---|---|---|---|---|---|---|---|
| 2 | 2 | 262.27 G | 3.24 M | 54.9 | 78.9 | 60.8 | 37.2 | 58.4 | 72.9 |
| 2 | 3 | 326.65 G | 3.62 M | 54.9 | 78.9 | 60.7 | 37.0 | 58.5 | 72.9 |
| 3 | 2 | 320.44 G | 4.03 M | 55.0 | 78.9 | 60.9 | 37.1 | 58.5 | 73.0 |
| 3 | 3 | 384.82 G | 4.40 M | 55.0 | 78.9 | 60.9 | 37.2 | 58.5 | 73.1 |
| 4 | 4 | 507.38 G | 5.56 M | 55.0 | 78.9 | 60.8 | 37.2 | 58.4 | 73.2 |

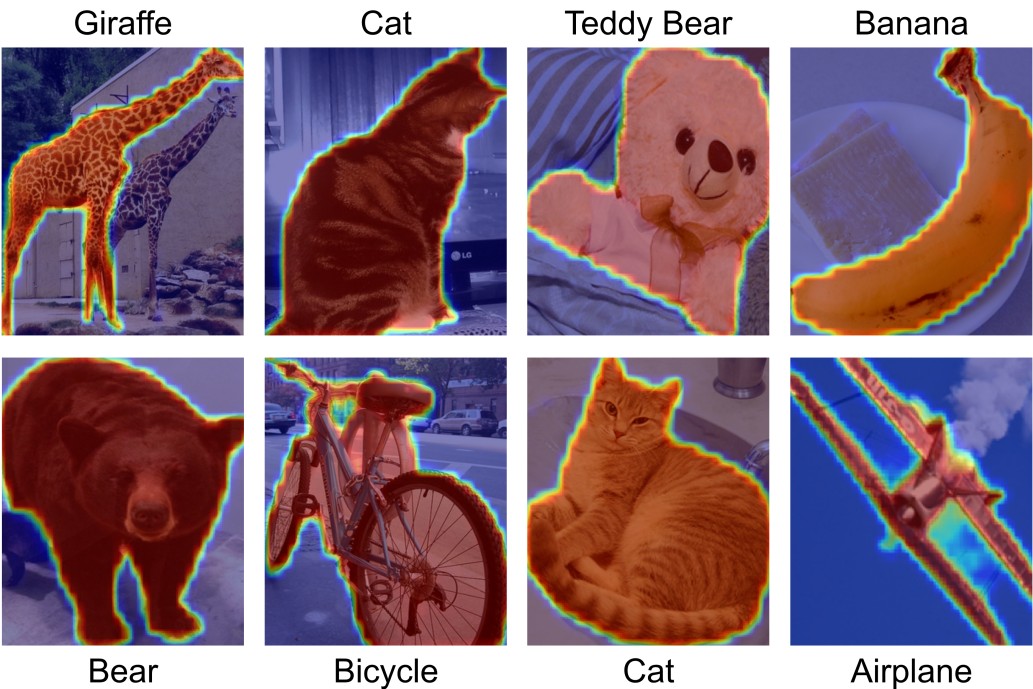

Figure 8: Visualizing the segmentation probability maps of our approach.

