# OpenReview forum: "Mask Frozen-DETR: High Quality Instance Segmentation with One GPU"
_ICLR.cc/2024/Conference — ICLR 2024 Conference Withdrawn Submission_

### Official Review · Reviewer_stNZ · 2023-10-27

**Soundness:** 3 good
**Presentation:** 3 good
**Contribution:** 2 fair
**Rating:** 5
**Confidence:** 4

**Summary:**

This paper studies how to efficiently insert a mask output head into a pretrained DETR object detector:  What layers can be left frozen, and how to insert extra adapters to make efficient use of the frozen features for the new task.  It finds that with a few well-placed attention/mapping layers, almost all of the new model can be left frozen, so that instance masks can be added with little training time compared to retraining all of the det+mask model from scratch.  The system is built incrementally, showing the effect of inserting adapters at different points, taking only those needed.  The final system achieves very good performance, recovering SOTA.

**Strengths:**

This is a clearly presented study of a kind of "transfer learning" with frozen pretrained model.  It's a little different from the most common transfer learning settings in that the training task (instance masks) is meant to supplement the original output (boxes), rather than replace it.  The ability of the system to do this with smaller amounts of compute is promising (see below).

**Weaknesses:**

This is a nice study of this sort of frozen model adaptation, but currently of limited use.  The setting uses COCO with all boxes and masks available, so there is little need to separate box and mask training: the data is fixed, and both box and mask labels come from the same source.  At best, one might say that optimizing box det first followed by masks later could make efficient use of training resources by allowing a (human or machine) model developer to focus on optimizing one task at a time, reducing the overall space of hyperparam variables and interactions --- but this isn't a point that was argued or supported.

A promising direction to push this further could be to see how well this method performs with different-sized subsamples of the mask data --- since mask labels are more expensive to gather than box labels, it is useful to use fewer of them, and collect just as many as needed in an iterative fashion.  This method may provide a simple means to do that while making efficient use of compute resources.  I think that would be an interesting application (particularly since the faster mask training could be used on incrementally larger datasets, as more mask labeling is performed).

Overall, this is a simple method whose main components are well documented and developed.  However, it hasn't yet been applied and evaluated in a context with clear benefit or advantages.

**Questions:**

Additional questions:

- comparison to SOTA methods in sec 4:  This appears to compare this method, which inits with an object detector already trained, to systems that jointly train object detection and masks from only a classification- or self-supervised pretrained backbone, without any obj det pretraining, unless I misunderstood.  I don't think this an appropriate comparison, as staged detection-first then mask training could also be done with any of these systems, freezing ether all or subsections of them as well.  IMO the apples-to-apples comparison would be to train the existing systems for obj det only, then add the mask outputs and train those with frozen features, possibly unfreezing some of the last layers or inserting trainable FFNs (but nothing more involved than that, as that would start to replicate this work)

- same for fig 1:  what are being trained in the comparisons -- masks only or everything?


- tables 2, 3:  would also be good to compare to FFN only or 2-hidden-layer FFN.  baseline qF has no ffn, only comparing directly the frozen features.  So it isn't clear how much of the improvement comes from just a FFN part, vs the attn/spatial combinations


- end of sec 3:  "additional enhancements ...".  while I appreciate these are in the appendix, I think more should be included in the main text, describing: (a) the impact / amount of improvement obtained from these enhancements, (b) whether they have been applied to all the models shown in the development (i.e. all frozen, + image feature encoder, + box encoder), only some, or none, (c) which of the other sota comparison systems use these or similar enhancements as well, and which do not but could still make use of them.


- I also didn't see a measurement (at least in the main text) of the inference-time cost of adding these extra layers, though that can be inferred approximately taking a fraction of the training-time flops.


- sec 3.3 channel mapper:  is the channel mapping layer applied to all dims, (h x w x d) -> d' (so 32*32*d -> d'), or just on the d dimension, preserving the h x w as (h x w x d) -> (h x w x d') ?

- table 4:  I think the number of hidden dims could go even lower --- what about going down to 8 (or even 1 or 2)?  There isn't any large performance drop-off yet in this table, while flops is still going down substantially.  same for table 10:  this could go lower than 16x16.

---

### Official Review · Reviewer_YtrW · 2023-10-29

**Soundness:** 2 fair
**Presentation:** 3 good
**Contribution:** 2 fair
**Rating:** 3
**Confidence:** 5

**Summary:**

The paper proposes an instance segmentation method based on the frozen DETR-based object detector. An additional mask network is used to predict instance masks within each bounding box. The paper is validated on the COCO dataset.

**Strengths:**

1.  The paper has a strait-forward motivation and idea. With the pre-trained DETR, only training the mask network is much faster and efficient, where the GPU hours in Table 6 supports the claim.

2. The paper writing is good and easy to understand.

3. The image feature encoder design is carefully studied in Table 2.

**Weaknesses:**

1. The paper has a low tech novelty. The mask head takes object features from RoI pooling using bounding box and object queries as input, which has been studied in queryinst. Also, it is similar to the mask head design in Mask R-CNN.

2. When comparing the GPU hours in Table 6, a more fair comparison should include (sum) both the training time for the frozen DETR and for the mask network. The training time of the object bounding box detector should be added.

3. Can the paper also provide inference speed comparison in Table 6 and Table 7?

4. Since the paper claims high-quality instance segmentation in the title, can the paper report the boundary AP? Can the paper provide more visual comparison cases with some high-quality instance segmentation methods, such as PointRend [a] or Mask Transfiner [b] or PatchDCT [c], or discuss the differences to them in the related works?

[a] PointRend: Image Segmentation as Rendering. CVPR,2020.
[b] Mask Transfiner for High-Quality Instance Segmentation. CVPR, 2022.
[c] Patch Refinement for High Quality Instance Segmentation. ICLR, 2023.

**Questions:**

1. Why Figure 4 is not mentioned in the paper?

2. With SAM/HQ-SAM, the instance segmentation can be directly done using the frozen DETR's predicted boxes as prompt. This requires no training at all. What's the clear advantage of the proposed one-gpu training method comparing to them? Can the author provide performance comparison with them using the same detector?

---

### Official Review · Reviewer_Nd3w · 2023-11-01

**Soundness:** 1 poor
**Presentation:** 1 poor
**Contribution:** 2 fair
**Rating:** 3
**Confidence:** 3

**Summary:**

The paper introduces a framework called Mask Frozen-DETR for instance segmentation, which utilizes a frozen DETR-based object detector to generate bounding boxes and trains a mask head to produce instance segmentation masks. The proposed method achieves SOTA results on the COCO dataset while being over 10 times faster to train.

The authors propose improvements in three main aspects: image feature encoder design, box region feature encoder design, and query feature encoder design. These enhancements improve the performance of instance segmentation based on a baseline setting.

**NOTE**: Overall, I personally think that the paper is very poorly written which makes it difficult to understand. For example, the variables in Equations 1-4 are not clearly explained and the connection to Figure 1 is also unclear. The authors describe their technical contributions in the subsequent sections (Sec. 3.2 - 3.4), but the unclarified variables and associated writing make it very difficult to understand these aspects. I believe the paper needs significant re-writing to make it understandable for any reader.

**Strengths:**

* The proposed approach leverages the strengths of existing object detection models and introduces an efficient instance segmentation head design.
* The experiments conducted in the paper showcase the quality of the proposed approach, achieving state-of-the-art results on the COCO dataset while significantly reducing training time.
* The significance of the paper lies in its ability to improve the efficiency of instance segmentation models by utilizing frozen object detector weights and introducing a lightweight mask decoder.
* The paper also investigates the impact of various enhancements, such as object-to-object attention, box-to-object attention mechanisms, mask loss on sampled pixel points, refined mask scoring scheme, and redesigned neck for backbone features.

**Weaknesses:**

The paper lacks clarity in explaining the proposed approach, making it difficult to understand the specific details of the Mask Frozen-DETR framework.

**Questions:**

-

---

### Official Review · Reviewer_kpyR · 2023-11-02

**Soundness:** 3 good
**Presentation:** 2 fair
**Contribution:** 3 good
**Rating:** 6
**Confidence:** 4

**Summary:**

The paper proposes Mask Frozen-DETR, an adapter architecture for instance segmentation. The main idea is to keep the original DETR-based detector frozen and only add a small number of parameters to adapt it to solve the instance segmentation (from object detection). Experiments are mainly conducted on COCO. Multiple DETR-based backbone detectors were used. Experimental results showed that the proposed method can achieve an AP comparable with current best method, i.e., Mask DINO, while reducing the training time by 10x. Written presentation is mostly clear and easy to follow.

**Strengths:**

- The idea of adapting DETR-based detectors for instance segmentation is interesting.
- The main benefit of the proposed approach is an adapter architecture for instance segmentation (from DETR-based detectors) that can reach to state-of-the-art level of Average Precision after 6 training epochs, thus reducing the training time by 10x (compared with Mask DINO).
- Various design choices and ablations to provide the readers good insights about the proposed architecture.

**Weaknesses:**

1. The proposed adapter works only with DTER-based detectors.

2. Written presentation could be improved:
- (This may be subjective) The Fig 3,5,6, and 7 have many things in common, they can be grouped into a large figure to simplify the presentation and save space. Doing so may also help to simplify the technical section 3, making it shorter and simpler.
- Figure 4 is never been referred from text.
- If more space is needed, ablation table 8, 9, 10 can report only the main metric of AP^{mask}.

3. Experiments are done on only 1 dataset.

**Questions:**

Please address the questions in weaknesses.